# Factors Associated with Bone Mineral Density and Bone Resorption Markers in Postmenopausal HIV-Infected Women on Antiretroviral Therapy: A Prospective Cohort Study

**DOI:** 10.3390/nu13062090

**Published:** 2021-06-18

**Authors:** Christa Ellis, Herculina S Kruger, Michelle Viljoen, Joel A Dave, Marlena C Kruger

**Affiliations:** 1Centre of Excellence for Nutrition, North-West University, Potchefstroom 2520, South Africa; salome.kruger@nwu.ac.za; 2Medical Research Council Hypertension and Cardiovascular Disease Research Unit, North-West University, Potchefstroom 2520, South Africa; 3Department of Pharmacology and Clinical Pharmacy, School of Pharmacy, University of the Western Cape, Bellville 7535, South Africa; mviljoen@uwc.ac.za; 4Division of Endocrinology, Department of Medicine, University of Cape Town, Cape Town 7535, South Africa; joeldave@endocrine.co.za; 5School of Health Sciences, Massey University, Palmerston North 0745, New Zealand; m.c.kruger@massey.ac.nz

**Keywords:** HIV, calcium, bone mineral density, black, postmenopausal

## Abstract

The study aimed to determine factors associated with changes in bone mineral density (BMD) and bone resorption markers over two years in black postmenopausal women living with human immunodeficiency virus (HIV) on antiretroviral therapy (ART). Women (*n* = 120) aged > 45 years were recruited from Potchefstroom, South Africa. Total lumbar spine and left femoral neck (LFN) BMD were measured with dual energy X-ray absorptiometry. Fasting serum C-Telopeptide of Type I collagen (CTx), vitamin D and parathyroid hormone were measured. Vitamin D insufficiency levels increased from 23% at baseline to 39% at follow up. In mixed linear models serum CTx showed no change from baseline to end (*p* = 0.363, effect size = 0.09). Total and LFN BMD increased significantly over two years, but effect sizes were small. No significant change in spine BMD over time was detected (*p* = 0.19, effect size = 0.02). Age was significantly positively associated with CTx over time, and negatively with total and LFN BMD. Physical activity (PA) was positively associated with LFN BMD (*p* = 0.008). Despite a decrease in serum vitamin D, BMD and CTx showed small or no changes over 2 years. Future studies should investigate PA interventions to maintain BMD in women living with HIV.

## 1. Introduction

Ethnic differences in fracture risk exist, with Caucasians being at the highest risk for fracture: i.e., several studies have shown that ethnicity could play a significant role in risk for fragility fractures. However, most of this data emanates from populations in North America and Europe. There are some data suggesting that black South African (SA) women are at lower risk for fractures than white South African women, thought to be due to their higher bone mineral density [1,2,3,4,5,6,7].

HIV-infected patients on antiretroviral therapy (ART) are experiencing an increased life expectancy [8,9]. With this, a variety of potential health consequences arises. Studies are showing the ubiquity of bone demineralization amongst HIV-infected patients [10,11,12,13,14]. The impact of HIV on bone health is considered to be complex and not yet fully understood. HIV infection as well as ART leads to increased bone loss [15]. A variety of factors contribute to the decreased bone mineral density (BMD) found within this group. These factors include traditional risk factors such as aging, a low body mass index (BMI), smoking, alcohol use and dietary deficiencies. Additional disease and treatment specific risk factors include chronic increased inflammation, immune reconstitution, opiate usage and decreased testosterone levels [16].

A recent meta-analysis including 35 articles, with 17 of the articles including fracture prevalence in HIV-infected patients and controls, indicated that HIV-infected patients had a 4.08% prevalence for all fractures and a 2.66% prevalence for fragility fractures [17]. HIV-infected patients were shown to have a higher prevalence for all fractures when compared to control groups [17].

Achieving an optimal dietary calcium intake is known to be beneficial for the attainment and maintenance of BMD [18]. However, Black SA women have a low calcium intake [19]. An optimal calcium intake during late adolescence and early adulthood is important as BMD peaks during these time periods [18]. When compared to the general population, HIV-infected patients have a 58% higher fracture rate [20]; this may be due to living longer with HIV and the cumulative exposure to ART with adverse effects on bone health [21,22]. The initiation of ART and the possible associated bone loss will result in an increase in the numbers of HIV-infected patients with decreased bone mass that may increase fracture risk in these patients [21,22,23,24]. There are limited guidelines aimed at preventing bone loss in people living with HIV (PLWH) [15].

There are sparse data from southern Africa on bone health in the HIV-infected patient population. The data that are available are mainly limited to younger PLWH [25]. There are no longitudinal studies of calcium intake and BMD in older HIV-infected women. Therefore, the aim of this study was to determine the factors associated with BMD and a bone resorption marker over a period of two years in black postmenopausal HIV-infected women on ART.

## 2. Materials and Methods

### 2.1. Study Population

The study participants were recruited from outpatient clinics in Potchefstroom in the North West Province, SA, where the majority of patients are of Tswana ethnicity (nwpg.gov.za/nwglance.htm. Accessed on 14 June 2021). According to the inclusion criteria, a total of 120 postmenopausal HIV-infected women, aged > 45 years on ART were enrolled. Exclusion criteria include use of anti-osteoporotic agents and drugs known to affect BMD (corticosteroids, thyroid medication, anti-vitamin K agents, diuretics, anti-epileptic drugs, β-blockers); secondary causes of osteoporosis (chronic liver disease, chronic obstructive pulmonary disease, chronic renal disease, immobility, rheumatoid arthritis, gastrectomy, malabsorption syndromes); diagnosed diabetes mellitus; history of metabolic bone disease; self-reported use of calcium and/or vitamin D supplementation, anti-acids containing calcium; high alcohol consumption (≥3 units/day) and a fracture within the last six months. Women with severely low BMD (T score < −3) at baseline could have been enrolled, but were excluded and referred for medical treatment after baseline measurements.

A power calculation was performed using G-Power version 3.1.9.2 to estimate the required sample size for this study. An estimated change in total BMD from 1.05 to 1.0 g/cm^2^ in the group (difference 0.05) and a standard deviation (SD) of 0.11 g/cm^2^ were used, with 80% power and a 5% level of significance. The estimated difference between the change in total body BMD and SD are based on existing data from a cross-sectional study in HIV-positive black women studied in 2014 in the North West Province [26]. The power calculation indicated that for group comparisons (lowest versus highest values of variables), 53 women per group would be sufficient to show a difference of 0.05 g/cm^2^.

### 2.2. Data Collection

Data was collected at baseline and at one and two years of follow-up. Baseline and follow-up visits were scheduled over time until all measurements had been completed at the metabolic unit of North-West University.

### 2.3. Measurements

Socio-demographic and health information of the women (age, educational status, housing, occupation, smoking, alcohol consumption, chronic medication) was obtained by an interviewer-administered structured questionnaire, used previously in a group with a similar age range in the prospective urban rural epidemiology (PURE) study [27]. Time period since diagnosis and since initiation of ART was also self reported by the participants. Anthropometric measurements (height, weight, waist circumference, mid-upper arm circumference, calf circumference) were performed by trained postgraduate Nutrition students using standard methods with participants wearing light clothing [28]. Height (cm) was measured to the nearest 0.1 cm with the participant barefoot and with the head in the Frankfort plane, and weight was measured to the nearest 0.01 kg on an electronic scale and stadiometer (Seca model 284, Hamburg, Germany). BMI was calculated as weight (kg) divided by height (m) squared [27]. Waist circumference was measured using a flexible steel tape (Lufkin, Cooper tools, Apex, NC, USA) around the waist at the level of minimal girth between the lower rib and iliac crest, to the nearest 0.1 cm. Mid-upper arm circumference (MUAC) was measured using a flexible steel tape to the nearest 0.1 cm on the participant’s left arm. Calf circumference was measured in a similar way at the medial surface, at the level of the largest circumference.

Dietary intake was assessed at baseline and at one-year follow-up by trained fieldworkers using a standardised quantitative food frequency questionnaire, a validated food portion photo book and food models to estimate portion sizes and convert intakes to weight [29,30,31,32]. Nutrient intake was calculated using the SA food composition database [33]. Physical activity (PA) was assessed at the same visits by a trained fieldworker using the Global Physical Activity Questionnaire (GPAQ) recommended by the World Health Organization (WHO) [34].

BMD was measured by a registered radiographer through dual energy x-ray absorptiometry (DXA) with the default Hologic settings (Hologic Discovery W, APEX system software version 2.3.1): total body, lumbar spine and left femoral neck (LFN) hip, at baseline, and at one- and two-year follow-up visits. The coefficient of variation (CV) of the femoral neck ranged between 1.1% and 1.2% and the CV for the lumbar spine BMD ranged between 0.7% and 0.9%. Verbal feedback was given to each participant at each measurement date. Participants were required to wear a cotton gown without any metal trimmings and all jewellery was removed.

Fracture risk was assessed using a checklist developed from information in the literature [35,36]. The checklist included eleven fracture risk factors: age > 40 years, female sex, underweight (BMI < 18.5 kg/m^2^), previous fractures, parent fractured hip, currently smoking, glucocorticoid therapy and proton pump inhibitor use during the last six months, rheumatoid arthritis, secondary osteoporosis and alcohol consumption of ≥3 units per day. Currently no validated fracture risk instrument is available for South African populations. The number of risk factors was recorded and represents the sum of the fracture risk conditions present in each participant.

### 2.4. Blood Analyses

All blood samples were collected in serum tubes and centrifuged as soon as possible after sample collection. Serum was prepared and stored in a bio freezer at −80 °C. Fasting serum concentrations of a bone resorption marker, C-Telopeptide of Type I collagen (CTx, B-CrossLaps/serum Elecsys Cobas e100), serum vitamin D (25(OH)D), Vitamin D total Elecsys Cobas) and parathyroid hormone (PTH, PTH Elecsys Cobas e100) were measured by immunoassay (Roche Diagnostics Pty (Ltd), Johannesburg, South Africa). All samples were analysed together in one batch with the same controls for each year of measurement. Serum 25(OH)D, PTH and CTx were measured at baseline, and again at one- and two-year follow-up visits. A serum 25(OH)D concentration of <20 ng/mL was used to classify vitamin D status as being deficient, 20–29 ng/mL as insufficient and ≥30 ng/mL as sufficient [37,38]. The intra-assay coefficients of variation (CV) were 3.8% for vitamin D, 1.2% for PTH and 2.4% for CTx. The inter-assay CVs were 16.3%, 8.1% and 10.3% respectively.

### 2.5. Statistical Methods

Data was tested for a normal distribution with the Shapiro–Wilk test and QQ plots. Descriptive statistics of socio-demographic data, dietary intakes, PA, BMD at the different sites (baseline), 25(OH)D, as well as PTH and the bone resorption marker CTx, were presented as means and SD (data with normal distribution) or median and interquartile range (non-normal distribution). Variables with a non-normal distribution were logarithmically transformed and the distribution of the transformed variables was assessed. Changes in the outcomes (BMD of the total body, lumbar spine and LFN of the hip, CTx and 25(OH)D from baseline to end were assessed by mixed linear models. Changes in PTH (non-normal distribution even after log transformation) were tested by the Wilcoxon signed rank test. Associations between variables were explored with Pearson (normal distribution) and Spearman correlation analysis (non-normal distribution). Differences between outcome variables of smokers and non-smokers were compared using the Mann–Whitney test.

Linear mixed models were used with CTx and BMD, respectively, as the dependent variables (outcomes) and dietary calcium intake as the primary exposure, with adjustment for variables identified to correlate with the exposure or outcomes (age, household income, education level, PA level). Separate models were used for whole body BMD, spine BMD and LFN BMD, respectively. Time was treated as a fixed effect and participant as random effect. The restricted maximum likelihood (REML) approach was used as a likelihood function calculated from the residuals obtained after adjusting for the fixed effects. The unstructured covariance structure was applied.

Statistical analyses were performed using the SPSS version 26 statistical software program. The level of significance was set at *p* ≤ 0.05.

## 3. Results

The demographic information of the participants is summarised in Table 1. Initially, a total of 120 postmenopausal women were included at baseline. After 1 year, drop outs were due to death (5), refusal to follow up (3) and untraceable (13). After 2 years, the drop outs were due to death (8), refusal to follow up (3) and untraceable (5), as indicated in Figure 1. Our participants had HIV infection for a mean duration of 10 years (±5.12) and received ART treatment for 9 years (±4.79)

The prevalence of vitamin D deficiency and insufficiency increased over the two years and median serum 25(OH)D concentrations decreased, whereas serum PTH showed a trend of an increase. Alcohol intake within the group was low. A small percentage of our group reported being current smokers (14%), with a median of 2 cigarettes per day. PA increased from baseline to 2 year follow up, but sedentary time remained consistent over time. The most common fracture risk factors at baseline were age and female sex (100% of participants), followed by previous fracture and currently smoking (both 14.2%). The highest number of risk factors present was five out of a maximum of 11 (5% of women). Six women developed incident fragility fractures at the 1-year follow-up and an additional four reported new fragility fractures at the 2-year follow-up. The proportion of current smokers decreased over the two years from 14.2% to 11.9%.

Spine, total and hip BMD, CTx, PTH and vitamin D showed no differences between smokers and non-smokers. Correlation analysis is presented in Table 2. There was a positive correlation between age and PTH (r = 0.20, *p* = 0.03), but no correlation between PTH, or 25(OH)D, respectively, with BMD at any site, or CTx. There was no correlation between the number of fracture risk variables present per participant and any of the BMD variables, or CTx.

In mixed linear models serum CTx (log transformed) showed no change from baseline to end (*p* = 0.363, effect size = 0.09). Total BMD increased significantly from baseline to the 1-year follow-up (*p* = 0.02), but not to the 2-year follow-up. The effect size of the change over time in total BMD was small (0.03). Similarly, LFN BMD increased significantly over two years, but with a small effect size (0.06), while there was no significant change in spine BMD over time (*p* = 0.19, effect size = 0.02). Serum 25(OH)D concentration showed a significant decline from baseline through to year 2 follow-up (*p* < 0.0001) with an effect size of 0.39. There were no significant changes in PTH from baseline over the two years (*p* = 0.08), but a clear trend of an increase over time (*p* = 0.08).

In subsequent linear mixed models, age was significantly positively associated with log CTx over time, and negatively with total and LFN BMD (Table 3). Total PA had a weak positive association with LFN BMD over time (β = 0.0003, *p* = 0.008). Calcium intake was not associated with any outcomes from baseline to end, with only spine BMD showing a weak trend of a positive association. The Bayesian information criterion (BIC) is indicated for the goodness of fit of the different models.

## 4. Discussion

The main findings of this study are that time, age, and PA are associated with BMD. Total BMD as well as LFN BMD increased significantly over the two year time period; however, the effect size remained small. Spine BMD and serum CTx showed no change over the two year time period. It is important to note that the effect size for changes in both total BMD and LFN BMD were small, and therefore probably not clinically relevant. We considered calcium intake, PA, age, educational status, serum vitamin D, PTH, alcohol intake, as well as smoking as possible factors that could be associated with BMD and a bone resorption maker in our study participants, but only age and PA were shown to have associations (with LFN BMD). A recent study contested the idea of genetic protection against osteoporosis at all skeletal sites in black SA women [7].

We found a small, but significant, increase in total BMD from baseline to 1-year follow-up, but this change was not maintained at the 2-year follow up. LFN BMD also showed a small increase over the two years, while spine BMD and serum CTx showed no change over the two years. It is important to note that the effect size for changes in both total BMD and LFN BMD were small, and therefore not clinically relevant. An important observation to make is that the decline in BMD was slow. The small changes in BMD at all sites must be seen in the context of decreased vitamin D levels, trend of increased PTH levels and a dietary calcium intake < 800 mg/day, all of which would have attenuated any potential increases in BMD. Hamill and co-workers [37] reported slightly higher BMD values in urban, black premenopausal HIV-positive women who were ART-naïve than what we found in our study (hip BMD = 0.988, LFN BMD = 0.923, lumbar spine BMD = 1.006). This difference could be attributed to the participants being younger in age (premenopausal). Another study found that low BMD was significantly more frequent in HIV-positive women compared to women without HIV [39]. Although there was no decrease in BMD over the two years of follow-up, 8.3% of the women reported incident fragility fractures over the two years of follow-up.

HIV infection, as well as ART are known to result in increased bone loss [40]. Mondy and co-workers [41] found that a decreased body weight, smoking and a long period of HIV infection (≥17 years) were primary factors associated with a low BMD. Our study participants reported a shorter mean duration of HIV infection (mean of 10 years) and therefore, it was possibly not long enough to show the same effect. Several studies have shown a high prevalence of low BMD in HIV-positive persons. These studies were, however, primarily based on participants with a lean or normal BMI [12,42,43]. The majority of our participants were overweight or obese, with no indications of weight loss over the three time-points of observation. This confirms the protective effect that body weight has on the prevention of bone loss. Conradie and co-workers (2014) [7] also found that the decreased loss of bone found in black women was related to a progressive increase in body weight in aging black women.

Biochemical markers of bone turnover are considered potential predictors of fractures. This is due to their interrelationship to bone quality as well as bone quantity [44,45]. Population-based studies have indicated that an increased bone turnover is associated with an increased risk of osteoporotic fractures. However, in our study, serum CTx showed no change over a period of two years. In the linear mixed models, we found that age was positively associated with log CTx over time, and negatively with total as well as LFN BMD. Higher levels of CTx were reported in a study conducted in non-HIV-infected black postmenopausal SA women than in the present study, and this was significantly associated with an increased loss of BMD over a two year period at the distal radius [46].

Bone turnover is either activated through mechanical stimuli on the bone or through systemic changes in homeostasis, resulting in the production of oestrogen and PTH [47]. PTH is considered to be necessary for the preservation of serum calcium concentrations [48]. The action of PTH on the osteoblastic lineage cells results in both the differentiation and the activation of osteoclasts, which leads to bone degradation as well as initiation of the resorption phase [49]. We found a trend of an increase in serum PTH over two years, in line with a significant decrease in serum 25(OH)D over the same time. This observation confirms the inverse relationship known to exist between vitamin D and PTH [50]. One to four percent of the variation found in PTH levels can be explained by vitamin D status [50,51]. Patients with secondary hyperparathyroidism are also known to have increased levels of PTH in response to low levels of serum 25(OH)D [37,52,53].

There was a gradual increase in the percentage body fat mass over the 2-year observational period. A large portion of the women included in this study were also classified as overweight according to their BMI. A recent study found that postmenopausal women with a higher BMI (>27 kg/m^2^) had lower levels of 25(OH)D [54]. This is similar to the results of our study, as serum 25(OH)D decreased over the observational period. This decrease in 25(OH)D concentration could be associated with the increase of fat mass.

Dietary calcium is known to play an important role in bone health [55,56]. Within this group, calcium intake was not significantly associated with any of the outcomes. The lack of association between variables could be an indication that our follow-up duration was not sufficient to detect any associations. Urbanisation is known to be associated with lower calcium intake in black SA women [57]. Furthermore, lower consumption of milk and dairy products was found in urban women when compared to rural women in a SA study [57]. Another possibility to take into account is that our group could have been under- or over-reporting on calcium-rich food intakes. A study conducted by Pedrera-Zamorano and co-workers [56] found that the protective effect of calcium on bone might possibly not be noticeable in postmenopausal women with a calcium intake lower than 800 mg/day. A study conducted in a similar population group also found participants to have low calcium intakes, with only 19.6% of the participants reaching an intake higher than 800 mg/day [58]. Ensuring a dairy intake of at least two cups per day (500 mL) has been shown to lower the incidence of osteoporosis as well as associated fractures [59].

PA showed a significant positive association with LFN BMD across the study duration (2 years). PA within this population group may reflect lifelong PA, which is unlikely to change or improve over time. The literature shows differing results regarding the association between PA and bone health. The discrepancies in results may be due to disparity in measurements of PA as well as population groups [60,61]. Studies in high risk population groups such as postmenopausal women have found that PA improved BMD as well as decreased fracture risk [62,63,64]. Limited data is available on the impact of PA on BMD in people living with HIV, however it is expected that PA will produce similar benefits [65,66,67]. A study conducted by Perazzo and co-workers [66] found that, in a group of relatively healthy HIV-infected participants on stable ART, those that reached the recommended level of PA, had increased BMD levels at the total hip in comparison to those not meeting this level of PA.

Limitations of this study include the observational design of the study and the lack of a HIV-uninfected control group, resulting in the inability to attribute causation. Participants originated from a segmented area in Potchefstroom, in the North West Province therefore extrapolation to other population groups should be carried out with caution. It is also important to note that the participant group for this study was small, which could have impacted overall results. Other limitations include the lack of information on years since menopause and that no bone formation markers, such as P1NP were measured due to lack of funding. Despite these limitations, our findings add to the existing body of evidence on calcium intake and serum vitamin D concentration in relation to changes in BMD over time within this population group.

## 5. Conclusions

In conclusion, this study found practically insignificant increases in total and LFN BMD over a 2-year follow up, while spine BMD and serum CTx showed no change over the two years. It could be beneficial to have a longer study duration to determine if any differences will appear in the outcomes. Future studies should also investigate the association between PA and LFN BMD and interventions to strengthen these outcomes.

## Figures and Tables

**Figure 1 nutrients-13-02090-f001:**
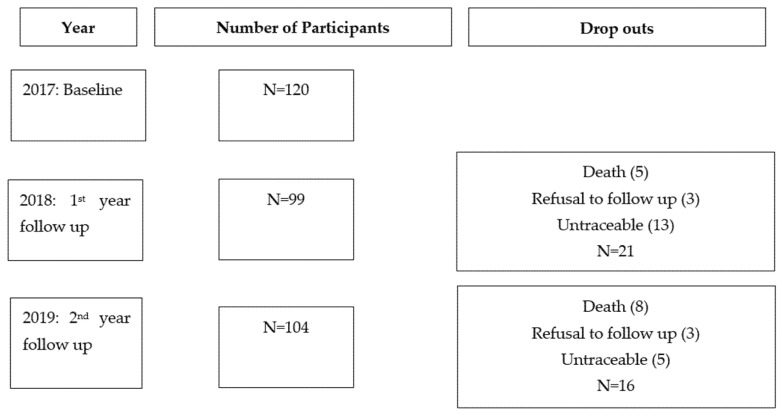
Number of participants per year and drop out reasons.

**Table 1 nutrients-13-02090-t001:** Study participants’ characteristics from baseline to 2-year follow-up.

Variable	Baseline (2017) (*n* = 120)	Year 1 (2018) (*n* = 99)	Year 2 (2019) (*n* = 104)	*p* * (for Outcome Variables Only)
	Median (IQR)	Median (IQR)	Median (IQR)	
Age (years)	50 (48–55)	51 (49–56)	52 (50–57)	
Weight (kg)	66.6 (54.1–80.0)	68.1 (57.9–83.0)	67.6 (54.8–83.9)	
Height (cm)	156.6 (151.5–161.0)	157.5 (152.3–161.5)	156.7 (151.8–160.4)	
BMI (kg/m^2^)	27.1 (22.4–32.6)	28.4 (23.5–32.2)	28.3 (23.0–33.2)	
Waist circumference (cm)	85.0 (76.0–97.3)	88.6 (76.9–96.6)	90.8 (76.7–101.0)	
Calf circumference (cm)	33.9 (30.0–37.5)	34.4 (30.7–37.6)	33.9 (30.2–37.7)	
Mid-upperarm circumference (cm)	31.6 (28.0–34.4)	31.6 (27.6–34.1)	31.3 (27.3–33.7)	
Total BMD (g/cm^2^)	1.04 (0.95–1.12)	1.04 (0.98–1.14)	1.03 (0.96–1.11)	0.02
Spine BMD (g/cm^2^)	0.84 (0.75–0.95)	0.88 (0.79–0.98)	0.87 (0.78–0.97)	0.19
LFN BMD (g/cm^2^)	0.74 (0.65–0.84)	0.73 (0.65–0.84)	0.76 (0.68–0.86)	0.0001
Lean mass (kg)	37.0 (32.9–42.6)	38.5 (33.8–45.2)	37.7 (34.0–45.6)	
Percent body fat mass (%)	38.6 (33.8–43.6)	39.1 (33.3–42.8)	39.9 (33.4–43.6)	
Serum 25(OH)D (ng/mL)	36.6 (28.1–44.2)	31.5 (25.0–41.1)	30.1 (24.9–37.6)	<0.0001
PTH (pg/ml)	41.3 (33.02–58.2)	42.9 (31.7–57.1)	47 (36.9–62)	0.08
CTx (ng/ml)	0.48 (0.35–0.75)	0.48 (0.35–0.65)	0.48 (0.34–0.69)	0.36
Dietary intake			-	
Ca-intake (mg/d)	668.3 (413.2–911.9)	517.6 (352.3–794.1)		
Energy (kJ)	10247 (7680–12514)	9330 (6166–11247)	-	
Fat (g)	76.6 (50.0–100.0)	57.1 (44.5–84.2)	-	
Protein (g)	75.6 (57.1–93.6)	68.9 (51.0, 96.5)	-	
Alcohol- intake (g/d)	0.0 (0.0–0.86)	0.0 (0.0–1.3)	-	
Physical activity (PA)				
PA (MET min/week)	2880 (960–7190)	2970 (840–6740)	3240 (2040–4530)	
Sedentary				
Time (min/day)	240 (150–375)	240 (150–375)	240 (180–360)	
	*n* (%)	*n* (%)	*n* (%)	
1st line ART (FTC; TDF; EFV)	105 (87.5)	81 (81.8)	84 (80.8)	
2nd line ART (AZT; 3TC; LPV/r)	15 (12.5%)	18 (18.2)	20 (19.2)	
Vitamin D status				
Deficiency	12 (10.2)	14 (14.3)	12 (11.5)	
Insufficiency	23 (19.5)	28 (28.6)	39 (37.5)	
Sufficiency	83 (70.3)	56 (57.1)	53 (51.0)	
Education				
No school & primary	44 (36.7)	29 (29.3)	31 (29.8)	
Grade 8–11	44 (36.7)	40 (40.4)	43 (41.3)	
Grade 12	31 (25.8)	29 (29.3)	29 (27.9)	
Tertiary	1 (0.8)	1 (1.0)	1 (1.0)	
Income (per month)				
None	6 (5.0)	3 (3.0)	10 (9.6)	
<R500–R1000	31 (12.5)	21 (21.2)	14 (13.5)	
R1001–R6000	74 (61.7)	64 (64.6)	72 (69.2)	
>R6000	9 (7.5)	11 (11.1)	7 (6.7)	
Hypertension	55 (45.8)	46 (46.5)	48 (46.1)	
Diabetes mellitus	10 (8.3)	9 (9.1)	8 (7.7)	

BMI = body mass index; BMD = body mass index; PTH = parathyroid hormone; CTx = C-Telopeptide of Type I collagen; Ca = calcium; MET = metabolic equivalent; 3TC (lamivudine); TDF (tenofovir); EFV (efavirenz) FTC (emtricitabine). * Changes from baseline to end determined by linear mixed modelling (in participants with data at all three data points).

**Table 2 nutrients-13-02090-t002:** Spearman and Pearson correlations between age and lifestyle and bone variables.

Variable	HH Income		PA		LFN BMD		Total Spine BMD		Total BMD		CTx	
Spearman correlations												
	r	*p*	r	*p*	r	*p*	r	*p*	r	*p*	r	*p*
Age	−0.03	0.711	−0.06	0.503	−0.38	<0.0001	−0.28	0.002	−0.41	<0.0001	0.30	0.001
Pearson correlations												
Serum vit D					0.001	0.990	−0.003	0.970	0.000	0.999		

HH = household; PA = physical activity; LFN = left femoral neck; BMD = bone mineral density; CTx = C-Telopeptide of Type 1 collagen.

**Table 3 nutrients-13-02090-t003:** The association between bone health variable and lifestyle and dietary factors (linear mixed models).

Parameter	Log CTx		Total BMD		Spine BMD		LFN BMD	
	Estimate	*p*	Estimate	*p*	Estimate	*p*	Estimate	*p*
Time	0.04	0.051	−0.02	0.003	−0.50	0.129	0.002	0.799
Age	0.01	0.009	−0.01	<0.0001	0.07	0.02	−0.01	<0.0001
Education	−0.02	0.45	0.009	0.501	0.12	0.509	0.003	0.792
PA	0.0003	0.24	0.0002	0.093	−0.0002	0.583	0.0003	0.008
Calcium	−0.0005	0.25	−0.0001	0.410	0.0007	0.123	−0.0004	0.770
Bayesian information criterion	5.39		−299.5		831.6		−274.1	

PA = physical activity; BMD = Bone mineral density; LFN = Left femoral neck; CTx = C-Telopeptide of Type 1 collagen.

## Data Availability

The data presented in this study are available in the article.

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
