# Peer review of "Factors Associated with Bone Mineral Density and Bone Resorption Markers in Postmenopausal HIV-Infected Women on Antiretroviral Therapy: A Prospective Cohort Study"

_nutrients, 2021, doi:10.3390/nu13062090_

Round 1

Reviewer 1 Report

This report describes a longitudinal, prospective study (with one-year follow-up) on the BMD variations in postmenopausal women affected by HIV. Although the subject of the study is not original, it is the first report of this type in postmenopausal black women with this disease.

Nonetheless, I have several comments, which follow below.

Introduction

The prevalence of fragility/osteoporotic fractures in patients with HIV have to be further  addressed and discussed (e.g. People with HIV infection had lower bone mineral density and increased fracture risk: a meta-analysis. Chang CJ, Chan YL, Pramukti I, Ko NY, Tai TW.Arch Osteoporos. 2021 Feb 27;16(1):47. doi: 10.1007/s11657-021-00903-y).

Method section

Variation Coefficient (CV) of the DEXA instrument has to be calculated and specified in the text (both for lunbar spine and femur), so that the changes in BMD can or cannot be considered significant (e.g. >2 CV).

Please, specify the intra and interassay CV of the included analytes.

Adopted definitions of vitamin D deficiency and insufficiency (along with appropriate references) have to be specified in the text.

Result and discussion section

History of previous, prevalent or incident fragility fractures have not been recorded: this is a major limit and these data should be collected and provided and correlated with BMD and the other parameters. Moreover, algorithm-derived fracture risk (e.g. by means of FRAX) should have been calculated.

Years since menopause should have greatly influence BMD. These data should have been included in the analysis.

Calcium intake decreased in the observation period: this is weird when and educational intervention is usually undertaken. This has to be commented in the discussion.

Reviewer 2 Report

The article is of potential interest related to changes in bone metabolism in women after meanopause however shows several limitations which require clear explanations.

1.First of all the title should be changed as the authors determined only one bone turnover marker, namely a  bone resorption marker CTX. My suggestion is „Factors associated with BMD and bone resorption in postemnopausal….”

Concentration  of 25(OH)D in the serum reflects vitamin D status of the individual but cannot be defined as a bone turnover marker.  Vitamin D is the key regulator of bone metabolism but is not listed among bone turnover markers, notably neither among bone formation markers ( which are for ex. P1NP, osteoclacin, BALP etc) nor among bone resorption markers (CTX, NTX, TrACP etc). Vitamin D deficiency contributes to higher level of parathyroid hormone, leading to the activation of bone turnover but has never been regarded by the experts in the field as a bone turnover marker.

2.Bone turnover may be assessed by bone formation and bone resorption markers. Both resorption and formation are  tightly coupled  processea and it would be interesting to know about the entire bone metabolism in HIV-infected women. Therefore,   the essential limitation of the submitted study is the lack of data on P1NP - a bone formtion marker which is available on the same instrument -Elecsys.

3.Further limitation of the study is that data on BMD and CTX in the non-infected postmenopausal women are not presented although probably such data for population of postmenopausal black women are available from the specific country registers. It is well known that BMD decreases in average from 0.5-1.5% per year in postmenopausal women  therefore  such comparison would be of  interest.

4.On page 3 line 127 the authors write „serum vitamin D (25OHD3), Vitamin D total Elecsys Cobas)”. The test for vitamin D total, correctly total 25(OH)D, is used for measurement of both forms 25(OH)D2 and 25(OH)D3, therefore   it is not possible to use the terms 25(OH)D3 and 25(OH)D interchangeably. In Table 2 the units for 25(OH)D , ng/mL, are missing.

5.Within 2 years of observation % of fat mass systematically increased and it is known that vitamin D is accumulated in adipose tissue therefore the decrease in 25(OH)D concentration could be due to the increase of fat mass. The authors should comment on this.

6.Finally the authors should clearly explain what their study adds to our current knowledge in the field.

Round 2

Reviewer 1 Report

The manuscript has improved and all the issues addressed. I have no further comments.

Reviewer 2 Report

The revised version of the manuscript has got substantially improved.